# Degradation of High Energy Materials Using Biological Reduction: A Rational Way to Reach Bioremediation

**DOI:** 10.3390/ijms20225556

**Published:** 2019-11-07

**Authors:** Stephanie Aguero, Raphaël Terreux

**Affiliations:** 1PRABI-LG—Tissue Biology and Therapeutic Engineering Laboratory (LBTI) UMR UCBL CNRS 5305, University of Lyon. 7 Passage du Vercors, CEDEX 07, 69367 Lyon, France; raphael.terreux@ibcp.fr; 2Pharmaceutical and biological Research Institute (ISPB), CEDEX 07, 69367 Lyon, France

**Keywords:** bioremediation, High Energy Molecules, HMX, protein design, molecular dynamics, nitroreductase, flavoprotein, substrate specificity

## Abstract

Explosives molecules have been widely used since World War II, leading to considerable contamination of soil and groundwater. Recently, bioremediation has emerged as an environmentally friendly approach to solve such contamination issues. However, the 1,3,5,7-tetranitro-1,3,5,7-tetrazocane (HMX) explosive, which has very low solubility in water, does not provide satisfying results with this approach. In this study, we used a rational design strategy for improving the specificity of the nitroreductase from *E. Cloacae* (PDB ID 5J8G) toward HMX. We used the Coupled Moves algorithm from Rosetta to redesign the active site around HMX. Molecular Dynamics (MD) simulations and affinity calculations allowed us to study the newly designed protein. Five mutations were performed. The designed nitroreductase has a better fit with HMX. We observed more H-bonds, which productively stabilized the HMX molecule for the mutant than for the wild type enzyme. Thus, HMX’s nitro groups are close enough to the reductive cofactor to enable a hydride transfer. Also, the HMX affinity for the designed enzyme is better than for the wild type. These results are encouraging. However, the total reduction reaction implies numerous HMX derivatives, and each of them has to be tested to check how far the reaction can’ go.

## 1. Introduction

High Energy Molecules (HEMs) is a term that stands for the class of materials known as explosives, propellants, and pyrotechnics. HEMs are required for a wide range of purposes in the fields of construction, engineering, mining, quarrying, space sciences (propellants), pyrotechnics, and currency production. They are also known for their military purposes [1]. The large-scale manufacturing and extensive use of HEMs for military purposes since World War II (WWII) have contributed to a high level of environmental pollution [2]. Contaminated sites are not easy to identify because they are not only located in present and former war zones but are also present among the military firing ranges; manufacturing, handling, and storage sites; and areas where they are used for industrial purposes [3]. In Australia, Canada, and the US, most of these sites have been located, and a minimal clean-up is in process. In Germany, the situation is more confusing because many of the explosive manufacturing facilities were demolished at the end of WWII. In other countries worldwide, the extent of contamination by explosives is either undetermined or not available to the public [4]. Moreover, some wars are still ongoing at present. Therefore, the environmental issue due to explosives remains a hot topic [5].

Decontamination solutions exist. The first treatment process in use was the incineration of explosive-contaminated soils. This method had the advantage of offering a high level of process control and efficient destruction and removal. However, burning was relatively expensive and also polluting due to the production of ashes. Then came biochemical solutions such as composting, the first biological treatment process to be tested and approved for military sites [4]. Composting requires the addition of bulking agents, increasing the volume of material. Unlike the ash created by incineration, composted material can support vegetation and is less expensive. Bioslurry is another soil treatment. In this process, contaminated soil is mixed with water and nutrients to create a slurry that can be combined in a bioreactor and treated with various bio-organisms. However, these treatments are ex situ and require additional costs regarding the equipment and process controls. A solution to ex situ bioremediation is the phytoremediation, which explores the ability of plants to remove pollutants from contaminated soils. The main limitations of this method are the toxicity of the contaminants: treatments are possible only if toxicity is not a factor with the candidate species. Moreover, absorbed pollutants and their metabolites must move from the bulk soil to the zone of influence near the roots for phytoremediation to occur. This requires a good solubility of the toxic waste. 

Among different forms of chemical explosives, 2,4,6-trinitrotoluene (TNT), hexahydro-1,3,5-trinitro-1,3,5-triazine (RDX), and 1,3,5,7-tetranitro-1,3,5,7-tetrazocane (HMX) are the most common (Figure 1). These explosives are highly stable compounds. They tend to blend with organic matter of soil and thereby to contaminate it [2]. Studies of toxicology on various organisms including bacteria, algae, plants, invertebrates, and mammals [6] have identified toxic and mutagenic effects of these common military explosives as well as their transformation products [4]. 

Research on TNT and RDX detoxification is still going on, and various phytoremediation-based approaches look encouraging. Recently, green grass was created to degrade TNT efficiently [7,8], and a field-applicable grass species capable of both RDX degradation and TNT detoxification has also been engineered [9]. Phytoremediation of TNT is very well studied, and the results look promising. However, phytoremediation remains more complex for RDX and even more so for HMX. HMX and RDX are nitramine compounds. Nitramines are less stable compared to aromatic nitro compounds such as TNT. Recently, several studies involving different microbes have been carried out to determine RDX degradation potential. There is, however, a lesser number of studies on aerobic and anaerobic microbial degradation of HMX. Despite being a close homolog to RDX, HMX shows more resistance to chemical and biological degradation than RDX, due to its very low solubility [10]. Recently, a study investigating the HMX degradation potential of the native bacterial isolate *Planomicrobium flavidum* strain S5-TSA-19 was conducted under aerobic conditions [11]. This bacteria strain showed efficient degradation, although some secondary metabolites (like methylenedinitramine and N-methyl-N,N′-dinitromethanediamine) formed during biodegradation of HMX are toxic or have unknown toxicity.

Over the past few years, the NAD(P)H-dependent bacterial nitroreductases (NRs) have received particular attention for their potential use in biodegradation and bioremediation of nitroaromatics [12]. NAD(P)H-dependent bacterial NR, also named nfsB enzymes, are capable of using either NADH or NADPH as reducing equivalent, in opposition to nfsA, which only uses NADPH. The nfsB enzymes are dimeric proteins and encompass two flavin mononucleotide cofactor (FMN). Both types of NR reduce a broad range of nitroaromatic substrates [13]. Reaction kinetic studies of NR have shown a simple ping-pong mechanism (Figure 2) without gating steps able to enforce specificity [14]. Electrons are transferred pairwise from the NADH cofactor to the oxidized flavin, and from reduced anionic flavin to nitroaromatics substrates [14,15] (Figure 2a,c).The substrate reduction has been proposed to occur via hydride transfer where two electrons and one proton are transferred in a common way [16]. It has also been proposed, in the case of NRs, that protons were transferred as solvent-derived protons via electron-coupled proton transfer [17]. The occurrence of a hydride transfer is dependent on the distance between the N5 atom of the reduced flavin and the atom donating or receiving the proton (Figure 2b). A distance of 3.8 Å between the two entities is optimal [18]. It has also been shown that upon reduction, FMN adopts a butterfly-like bending of the isoalloxazine ring system [19].

The reduction of nitro groups usually leads to hydroxylamines (Figure 2) and requires the transfer of four electrons. The two-electron nitroso reduction intermediate is not observed because the second two-electron reaction has a much faster rate than the first two-electron transfer [20,21]. It has been shown that the well-characterized NRs from *Escherichia Coli* and *Enterobacter Cloacae* (EcNfsB and EntNfsB, respectively) were able to reduce nitroaromatics into the corresponding hydroxylamines, but not into the amines [14,15,16,17,18,19,20,21]. However, aromatic reduction to amine has been observed: two NRs from *B. subtilis* [22] and more recently one NR from *Gluconobacter oxydans* 621H reduce TNT into the corresponding amino products [23]. The advantage of using these enzymes, apart from their ability to reduce a broad range of nitro compounds including the explosive TNT, is that the reduction of the nitro group into amine buries any toxicity issues. 

Miller et al. [24] have established a correlation between amine production and substrate properties, indicating that the best choices will have large pi systems and electron-withdrawing substituents. Likewise, they have shown that electron-withdrawing groups favor the reduction of nitro substrates. Thus, they suggest that smaller nitroaromatics will be less likely to undergo full reduction. This is why compounds as TNT undergo reductions to the corresponding amine. HMX has four nitro groups, which are electron withdrawing, but doesn’t get a pi system. The pi system allows for pi stacking between the substrates and the flavin, stabilizing the complex. Our challenge was to create a stable complex without the solicitation of pi stacking.

We focused our research on the NAD(P)H-dependent bacterial NR from *Enterobacter Cloacae* [25]. The NO2 > NO reduction reaction has been observed in this NR on a similar substrate, the nitramine RDX [13]. However, HMX is relatively insoluble in water, and much more stable than RDX. Its metabolization by the same type of enzyme (for denitrification purposes) is therefore more complicated. Residues 1-nitroso, 2-nitroso, and 3-nitroso HMX have already been observed, but the rate of the reaction is lower than those observed for RDX. Thus, we computationally modified the structure of the protein to generate a substrate specificity toward HMX. We worked on the PDB structure 5J8G [26] and rationally redesigned the active site of the NR on its reduced form by using protein design methods (Figure 3). Then the mutated structure was pushed through molecular simulation to evaluate the stability of the newly designed active site. Docking of cofactor confirmed that the performed mutations did not alter the bonding of the FMN electron donor NAD(P)H. All the subtlety of this work was to improve the specificity of an unspecific enzyme and to allow for the long-term reduction of non-aromatic nitrated substrates into amine.

## 2. Results

### 2.1. Design of the Active Site 

The native structure 5J8G shows the familiar symmetric dimeric fold of NRs [15]. The protein is composed of two intricated monomers. The H7 helix is central to the dimer interface. This configuration stabilizes the dimer even when bound to FMN or to various substrates [26]. The active site is a cleft at the protein surface. It is limited by helix 6 (amino acid 109–129) and 7 (134–148), strand 3, and helix 8 (156–175).

Multiple binding orientations have been observed in NfsB’s active site, based on the crystal structure of the oxidized enzyme. However, there is a lack of information regarding the substrate binding to the reduced form of the NR. Drigger et al. [27] have shown that, for the SsuE FMN reductase, a protein related to the flavodoxin-like superfamily, as well as the NR, a stacking of the NADPH on top of the flavin was a non-productive mode of binding. Thus, a productive binding would be related to the possibility of a hydride transfer. This transfer depends on the distance between the N5 atom of the flavin and the atom donating or receiving the proton, with a distance of 3.8 Å being optimal. Thus, we decided to position one of the nitro groups of HMX close to the hydrogen bound to N5 at a distance inferior to 3.8 Å. Then, we selected, around the HMX molecule, every amino acid within a distance of 4.5 Å presenting a lateral chain facing the active site. From this ensemble of amino acids, we excluded Glu165 and Gly166, which support efficient hydride transfer from NADH to FMN. We also excluded Asn71, Thr41, and Phe124, which are involved in interactions between NADH and FMN [15,16,17,18,19,20,21,22,23,24,25,26,27,28]. Phe124 was also excluded, as the amino acid was identified as conferring NR activity [29]. Moreover, the side chain of Phe124 interacts with the nicotinamide ring of NADP [26]. These particularities are shared with several other NR homologs [17,18,19,20,21,22,23,24,25,26,27,28,29,30].

As a final verification, the remaining residues were cross-checked against a sequence alignment clustering the 50 most similar protein sequences in the UniProt Database filtered according to a UniRef90 BLAST Search. Highly conserved residues across the alignment were excluded. We finally allowed the modification of Tyr68, Phe 70, Glu120, Tyr123, and Ala125 into every natural amino acid (Figure 4). A restrained energy minimization was then performed to relax the structure in order to keep the HMX molecule in a relevant position.

Then, we performed several trials using the Coupled Moves algorithm, using 30 identical starting structures (see Appendix A for script and param files). A total of 10,000 steps per structure was conducted. Residues Tyr68, Phe70, Glu120, Tyr123, and Ala125 were allowed to be mutated into any natural amino acid, including their associated side chain rotamers. The selection of amino acid identity and rotamer at each step was based on the calculated ROSETTA energy scores and chosen according to Boltzmann weighted probability. The simulation produced 6093 low-energy sequences. The results of the trials are illustrated under a logo sequence (Figure 5). The logo depicts the amino acid conservation among the sequences previously generated. 

As shown in Figure 5, the occurrence of each amino acid within the sequences is depicted by the total height of the letter. Biggest letters are identified as beneficial. Mutations G120D, A125G, Y68F, and F70E were accordingly proposed to improve the stability of HMX in the active site. Mutations of Y123 do not demonstrate clear results. The most frequently proposed mutation is the replacement of the tyrosine by an arginine. Mutations G120D, Y123K, and A125G are located within the H6 helix. It has been suggested that the substrate specificity was due to the plasticity of this helix. Indeed, H6 shows an elevated variability in amino acid and position for accommodating substrates of different sizes. Mutations Y68F and F70E are located on H4.

When compared to p-NBA (the bound substrate of the NR in 5J8G crystal structure used for mechanistic studies by Pitsawong et al. [26]), HMX shows a completely different structure: p-NBA is planar, allowing binding via H-bonds and pi staking above the re face of the FMN, as for the other substrates and analogs in the active sites of the NR and NfsB. However, the placement of p-NBA is not optimal for reactivity, as the nitro group to be reduced is too far from the reduction center (N5 of FMN). HMX does not have an aromatic structure. It consists of an eight-membered ring of alternated carbon and nitrogen atoms, with a nitro group attached to each nitrogen atom. This means that the molecule is not planar and can adopt four different crystalline conformations (alpha, beta, delta, and gamma). Stabilizations by pi stacking interaction are no longer possible. Transformation of one HMX conformation to another only occurs at high temperature. However, here, we consider the four conformations as only one possible substrate with several conformers for simplicity.

As shown in Figure 6b for the mutant NR, K14 forms H-bonds bridges between the protein, the FMNH2 cofactor, and HMX. This bridge maintains FMNH2 and HMX in close contact together. Additionally, there is an H-bond between non-cyclic part of FMNH2 and one nitro group of HMX. Finally, HMX is stabilized in the active site because of the electrostatic repulsion of the mutated G120D, which interacts with the hydrogens related to the carbon ring atoms. The rest of the molecule is exposed to the solvent. The residues Phe124, Thr41, Glu70, and Asn71 exhibit a side chain and a backbone segment, which are both very highly solvent exposed in the active site when HMX is not present. The presence of the ligand greatly reduces the solvent accessible surface area. HMX fits in a more homogeneous way in the active site of the mutant. Indeed, the WT NR stabilizes HMX with Asn71 and Thr 41, but also through Gly166 and Glu165 H-bonds, in a way similar to the binding mode of NADH to the NR (Figure 6a). The mutation G120D reduces the inner volume of the active site and allows for a better fit of HMX. In the mutant, one of the nitro groups of HMX is close enough to the N5 atom (4.35 Å) to allow hydride transfer. This first reduction step of HMX has been experimentally observed [31]. This situation is not surprising. However, the design of the active site aims to allow for both the experimental first step of the reaction and for the entire reduction of the nitro group.

After the design phase of the NR being completed (Figure 7), we needed to explore the time dimension parameter of the 3D NR model. Thus, the behavior of HMX in the newly design active site was investigated. The objective was to see if the generated structure could reach an energetic balance and, if yes, in which way.

### 2.2. Molecular Dynamic Simulations 

#### 2.2.1. Global MD Analysis

A first MD simulation of 130 ns was performed on the NR in a ligand-bound complex with p-NBA (PDB ID: 5J8G). Pitsawong et al. [26] showed that the NR does not generate paminobenzoic acid and thus does not appear to reduce nitro groups into amines [14,15,16,17,18,19,20]. p-NBA has been co-crystallized with the NR from Enterobacter Cloacae. We used this crystal as a starting structure to understand how p-NBA was stabilized in the active site. The dynamics show that the p-NBA is not stable in the active site during the whole dynamic but moves in and out of the pocket. When in the pocket, the aromatic ring of the p-NBA is stabilized against the flavin ring, probably involved in pi stacking interactions. The nitro group is oriented in a non-productive way, meaning that the nitro group is too far from the N5 atom of the flavin to trigger a hydride transfer. These observations are consistent with the crystal description of Pitsawong et al. Affinity calculation of p-NBA/NR interaction were calculated (Table 1).

A similar molecular dynamic simulation of 130 ns was performed on the Wild Type NR in a ligand-bound complex with HMX. We used the 5J8G crystal structure as a starting point. The p-NBA was replaced with HMX positioned in a productive way, according to the original p-NBA orientation, with one of its nitro groups close to the N5 atom of the flavin at a distance of 3,8 Å.

We observe that HMX doesn’t reach a stable position within the active site. It rolls in the space delimited by H6 and FMNH2 and struggles to find a stable position. As a consequence, none of the four groups is stabilized long enough within a radius of 3.8 Å around the reductive N5 of FMNH2 to consider a hydride transfer. Phe124 is positioned under one of the nitro groups and does not manage to stabilize it. Phe124 also cannot prevent HMX from leaving the active site, and HMX ends up escaping after 70.4 ns (frame 352) in one out of three simulations. During this particular dynamic, Tyr 123 failed to catch and get HMX into the site.

As a result, the WT NR does not successfully stabilize HMX because these weak interactions are not persistent in time and cause HMX to leave.

We did not use the docking method to place HMX in the NR active site because we had to dock the HMX between FMNH2 and H6. However, H6 is greatly variable in terms of position [15]. Such a variability, in terms of plasticity, may be directly related to the wide range of substrate accepted by the NR. Thus, while moving, the helix allows for the accommodation of more or less bulky substrates. The use of docking, even with induced-fit features, would not have given good results. By considering the helix as non-flexible, the pose wouldn’t have reflected the reality of the NR flexibility. By using our method, thanks to the dynamics, the helix rearrangement can be considered as accommodating our large substrate, while respecting our positioning constraint.

Another 130 ns MD simulation was performed on the mutant NR previously designed, in a ligand-bound complex with HMX. The input .pdb structure was directly extracted from the coupled trials after the selection of the lower energy mutant. This mutant of Enterobacter Cloacae NR shows the following mutations: G120D, Y123K, A125G, Y68F, and F70E. Throughout the simulation, HMX remains trapped in the cleft formed by the active site. This cleft is delimited by Phe124 upstream. Laterally, Lys123 blocks HMX and prevents it from leaving due to H-bonds connections through the amine portion of its lateral chain and the hydrogens connected to the carbon atoms of the HMX cycle. Alternatively, it also establishes H-bonds with the oxygen of the nitro groups. Downstream, Asn117 also makes H-bonds through its amino lateral chain. Phe68 occupies steric space above HMX, and places it against FMNH2. At no point during the dynamics does HMX go out of the molecule. It is stabilized in a relative way since it is able to roll in the previously described space. Thus, the nitro group presented at the reducing center N5 is never the same. This is partly due to conformational changes in the HMX.

#### 2.2.2. Stability Studies 

RMSD gives an overall picture of how much the protein structure has changed throughout the simulation. It provides an overview of protein stability over time. We captured RMSD of the protein, of the cofactor FMNH2 and each substrate (p-NBA and HMX). RMSD plots for all simulations are shown in Figure 8.

The first simulation captures the movements of the NR is complex with p-NBA, the initial structure of the NR co-crystallized acid para nitrobenzoic (Figure 8.1). All along the 130 ns of simulations, the WT NR protein is stable as shown by its low RMSD (2.2 Å). FMNH2 shows more movement relative to the protein (displacement of 3.5 ± 1.00 Å). This is because the cofactor is not covalently bound to the NR protein but through H bonds. This mode of binding allows for a certain flexibility of the structure, depending on the inner movements of the protein backbone and through lateral chains. p-NBA is not stable in the active site as it moves in and out. RMSD translates these movements by showing a broad variability.

In comparison, the WT protein in complex with HMX shows similar stability (Figure 8.2). The protein exhibits a RMSD value of 2.00 ± 0.15 Å, whereas the FMNH2 gets a higher RMSD value at 3.25 ± 1.50 Å. However, HMX goes out of the active site in one simulation, but not in the two others. In the first one, HMX exhibits a RMSD value of 18.73 ± 1.19 Å. It seems to be stable in the first few ns but finally goes out, to the same extent as p-NBA. HMX gets lowers RMSD values for the two other simulations (respectively 8.52 ± 3.57 Å and 7.21 ± 1.19 Å). HMX behavior is not similar among the three MDs. In the first one, residues F124 and K123 of H6 do not manage to lock HMX in the pocket. As a result, the molecule leaves the active site. In the two other ones, due to a slightly different orientation of their lateral chain, residues F124 and Y123 prevent HMX from escaping the active site. As a consequence, HMX manages to establish transient H bonds with K41. However, these H bonds are not sufficient to stabilize HMX, which rolls in the pocket.

Our design project aimed to create an NR able to stabilize and thus make possible the reduction of HMX nitro groups. The designed NR displays comparable stability (Figure 8.3) when compared to the WT NR (2.25 ± 3.05 Å for the protein, 3.25 ± 0.75 Å for the cofactor). HMX reaches a stable state when complexed with the mutant enzyme. Indeed, even if some artifacts are observed due to the periodicity of the solvation box, the global RMSD is quite stable, regarding the complexed WT NR: 6.41 ± 1.20 Å, 4.35 ± 1.97 Å, and 5.11 ± 1.76 Å for the triplicate MD. In the mutant NR, HMX is more stable in the active site and is adequately positioned. The mutations Y123K and G120D (H6) stick HMX against the re face of the flavin, establishing H bonds with the nitro groups of HMX. Y123K mutation plays a crucial role in maintaining HMX in the active site. In the WT NR, Y123 did not manage to stabilize HMX through H bonds. As a consequence, HMX escapes or is pushed in the pocket without reaching a stable state. In the mutant NR, K123 catches and stabilizes HMX through H bonding when the molecule moves away from the active site. G120D mutation also provides better stabilization through H bonds between the nitro groups of HMX and its two oxygens. F124 prevents HMX from leaving to the same extent as in the WT NR. Moreover, this optimal support is reinforced by K41, which forms H bond bridges between the protein, the FMNH2 cofactor, and HMX. Finally, HMX is stabilized at the required distance to observe the hydride transfer.

As a final verification, an additional simulation was launched to verify whether the designed active site conserves its stability in the absence of HMX. As shown in Figure 8(4), both FMNH2 and the protein maintain their stability with an RMSD even more stable than those with HMX (respectively (3.88 ± 0.38 Å for the protein and 1.88 ± 1.23 Å for the cofactor). The flat RMSD means that the designed protein is stable, with and without HMX bound to the active site. The absence of high variations may be due to the lack of HMX above H6, which is known to show a high flexibility to accommodate different substrates.

#### 2.2.3. Affinity Studies

Once the MD simulations of ligand recognition upon binding of HMX to the NR were performed, we also calculated the ligand-binding affinity.

The calculated binding free energies of each substrate for the WT and mutant NR were computed using the ensemble-average molecular mechanics energies combined with the generalized born and surface area continuum solvation (MM/GBSA) rescoring. MM/GBSA are popular methods used to estimate binding energies of small ligands to biological macromolecules. They are based on molecular dynamics simulations of the receptor–ligand complex. Each calculated binding free energy is averaged from snapshots extracted from the last four ns MD trajectories. The results are shown in Table 1.

We observed that the binding free affinity of the mutant NR and HMX complex is lower than those of the WT NR/HMX complex. This affinity value is also stronger than the WT NR/p-NBA one. These results tend to ensure that the designed enzyme complexed with HMX offers better stability compared to the WT NR. These calculations are confirmed by the structural study of the complex interaction.

#### 2.2.4. HMX Behavior during the Simulation

The distance between each oxygen from the HMX nitro groups and the reductive N5 of FMNH2 has been calculated during the 130 ns simulation. As previously described, HMX does not stay bound in the active site of both WT and mutated NR. While HMX leaves the WT NR after several rolls in the active site, it remains trapped in the mutated NR active site. However, it does not keep a stable position and also rolls under FMNH2, exposing different nitro groups to the reductive N5.

A graph showing the medium distance between each oxygen from HMX (O1, O2…O8) for the complexes WT NR/HMX and mutant NR/HMX is shown in Figure 9. As previously mentioned, the possibility of hydride transfer depends on the distance between the flavin N5 atom and the atom donating or receiving the hydrogen, 3.8 Å being optimal [18].

In the WT NR (Figure 9a), the distance to observe the reductive reaction is only inferior to 4 Å during the start of simulation. However, this distance is almost always inferior to this value for our mutant NR (Figure 9b).

These observations, combined with the affinity calculations, tend to prove that the mutant NR/HMX complex is more stable and that there is a higher probability of observing a hydride transfer between the mutant NR and HMX than between the WT NR and HMX.

### 2.3. Docking Studies

Our results show that the designed enzyme is able to better stabilize HMX than the WT NR, and that the affinity between the mutated NR and HMX is stronger than for the WT NR/HMX complex.

However, a specific point still needs to be clarified: is the mutant NR flavin still able to be reduced by NADPH? To answer this question, *in silico* docking of the NADPH co-factor to the predicted mutant NR structure was performed.

The last frame of the MD simulation for the mutant NR/HMX complex was extracted as .pdb. Prior to the docking, HMX, water molecules, and ions were removed. The docking zone was defined as follows: the FMN cofactor, H5 and H6 (residues 94 to 129), the loop between H3 and H4 (residues 67 to 74), the loop from amino acid 40 to 43, H7 (residues 138 to 142), the loop between H1 and H2 (residues nine to 22), and the loop between strand4 and H9 (residues 197 to 207). This zone spatially defines the active site. It also includes amino acids known to be involved in weak and dominant interactions with the NAPDH analog nicotinic acid adenine dinucleotide (NAAD). Indeed, Pitsawong et al, 2017 have shown that the nicotinic acid ring of NAAD stacks against the re face of the flavin over the uracyl and diazabenzene. They have also shown that weak interactions with the backbone of Gly120 and Thr67, and with the side chain of Asn71, engage the ribose hydroxyls. The dominant interactions stabilizing the phosphates are observed with the conserved side chains of Lys14 and Lys74. Also, the side chain of Phe124.B interacts with the nicotinamide.

A total of 150 docking poses were generated using the triangle match placement method associated to the London dG scoring function. Then, 25 top scoring poses were refined using the induced fit refinement scoring function GBVI/WSA dG with the generalized born solvation model (GBVI).

The top five scoring poses were evaluated. Three conformations of NADPH bound into the active site of the protein and interacting with FMN were observed. The ranking was based on the GBVI/WSA score and the maximum number of favorable non-bonded interactions between the mutant protein and NADPH. Configuration number 4 had the maximum number of protein–substrate–cofactor interactions and was hence considered to mimic the actual protein–ligand complex: carboxyl group of His11 forms one H-bond bridge between the protein, the phosphate group on NADPH, and the cofactor FMN atom 02 (Figure 10). The nicotinamide is not involved in pi stacking interactions with FMN, but it sits deep in the pocket, which contains the flavin rings, and which is stacked against the re face of the flavin over the uracyl and diazabenzene rings. Our observations are in good agreement with the crystal structure 5J8D of the NR bound in a complex with NAAD, an analog of NADPH [26]. The distance between the transferable hydrogen and the N5 of the flavin is 2.88 Å, allowing for a hydride transfer. 

## 3. Discussion

In this study, we designed the NR from *Enterobacter cloacae* to facilitate the reduction of a specific substrate: the high energy molecule HMX. This design is more likely a redesign of the active site, since only a few positions were mutated to allow a better stabilization of the ligand in the pocket. Indeed, in vitro studies have shown that HMX does not undergo full reduction. Different causes are possible—HMX has a very low solubility, and the NR is a soluble enzyme. Also, nitroaromatics are more likely to be stabilized in the active site, thanks to the pi stacking interactions with FMN, which is not the case of HMX.

The challenge here was to use a different way to stabilize HMX. The design allowed for the creation of various H-bonds with HMX (Figure 6), and the ligand fits more homogeneously in the mutant active site than in the WT NR. However, the model has been designed using a relatively rigid structure as a starting point. This is why we used MD simulations to allow for the relaxation of the complex and affinity calculations. RMSD calculations showed that the designed NR has a stability comparable to the WT NR. However, HMX reaches a more stable state when complexed with the mutant enzyme (Figure 9). Affinity calculations also indicate an improved affinity. As shown in Table 1, HMX and p-NBA contain oxygens with partially negative charges, suggesting electrostatic interactions, which should be a critical factor in the binding affinity. HMX has four nitro groups. Each of them has two H-bonds acceptors. As H-bonds and pi stacking are weak interactions, the presence of eight oxygens could compensate for the absence of the aromatic group and the subsequent absence of pi stacking for HMX. This also could explain the remaining of HMX in the WT NR active site during the start of the simulation. The escape of the ligand outside the pocket could be due to the size of HMX and the nature of the molecule, which presents inner movements, resulting in a larger occupied space than the one held by p-NBA.

This work also allowed us to show that the mutated NR is able to use NADPH in conjunction with oxidized FMN. In this regard, we performed a docking study. The optimal pose was selected among the top five scoring poses, according to the maximum number of protein-substrate-cofactor interactions. It has been shown that Asn71, Lys14, and Lys74 make polar contacts with the sugar-phosphate portion of the bound nicotinic acid adenine dinucleotide (NAAD), a precursor of NAD and NADPH [26]. It has also been shown that the NAAD binds the NR in an extended conformation [32] with its nicotinic ring against the face of the flavin. This is consistent with observations in other flavoenzymes [33]. These residues, identical or similar in NR homologs, suggest that NADH binding is similar among the family members. As none of the three conserved residues have been mutated during the design phase, we suppose that NADPH could adopt a stable conformation in the designed active site. However, our selected docking pose does not show the same interactions. NADPH is stabilized by a bond bridge between His 11, the phosphate group of NADPH, and the FMN atom 02. This configuration locks NADPH in the pocket. Also, we did not observe the interaction between the nicotinamide ring of NADPH and the backbone of Phe124, which was identified as conferring NR activity [34]. However, we observe the same position of the nicotinamide stacked against the re face of the flavin. Regarding the reaction, the hydride transfer is possible if the C4 of NADPH is 3.0Å from the flavin N5. This position promotes orbital overlap between the nicotinamide C4 hydrogen of NADH and the N5 of the isoalloxazine ring [18]. The selected pose of our study shows a distance of 2.88 Å between the FMN N5 and the hydrogen donor, which allows for the hydride transfer. Consequently, MD simulations would be relevant, as the structure, even minimized, could be trapped in local energy minimum. Indeed, docking was performed with an induced fit model could observe a rearrangement of the NADPH after a relaxation phase. 

One last question still needs to be answered: does the mutant NR accommodate intermediate HMX derivative structures? Our work aimed to design an enzyme able to reduce the nitro groups of the explosive HMX specifically. In the process, the design was performed with HMX as a starting substrate. However, HMX contains four nitro groups, and the affinity of the intermediate structures, shown in Table 2, must be studied. The point is to understand how far the reduction could go, and how much we could detoxify the molecule. In 2014, Pitsawong et al. [14] showed that the WT NR does not generate p-aminobenzoic acid from p-NBA and therefore appears to not reduce nitro groups into the corresponding amine. The chemical explanation of this limitation was brought by McCormick et al. [35], who showed that the reduction rate of the nitro group increases with the groups present in the para position (with the following priorities: NH2 <OH <H <CH3 <COOH <NO2). It could be interesting to evaluate whether our mutant NR could overcome this condition.

The interest in going beyond the hydroxylamine state lies in the fact that HMX derivatives are also considered toxic. Indeed, we understand that the toxicity of HEMs is highly connected to the presence of the nitro groups. However, the hydroxylamine derivative is also known to interact with biomolecules, including DNA, and thus causing toxic and mutagenic effects. The toxic effects are related to the electrophilic character of these derivatives, whereas the mutagenic effects are mainly due to the formation of hydroxylamine moiety adducts through esterification with guanine [36]. This is why it would be interesting to perform a docking study of the HMX derivatives on the designed NR to investigate if the docking poses could promote hydride transfer. Also, these results would have to be confirmed by MD simulations and affinity calculations to confirm the stability of these molecules in the active site.

From a technical point of view, it is important to address two points. Regarding the design phase, the choice of the algorithm depends on the protein design problem: given a desired structure, can we design an amino acid sequence capable of assuming a target structure? The goal of a protein design algorithm is to search for all the possible conformations of a sequence that could match a target fold. Then it must rank the sequences accordingly to the lowest energy conformation of each one, as determined by a protein design energy function. This ranking is highly connected to the optimization problem consisting of finding the conformation of minimum energy. A bunch of algorithms have been developed to solve the protein design problem. They are divided into two broad classes—exact algorithms, such as dead-end elimination (DEE), that lack runtime guarantees but guarantee the quality of the solution; and heuristic algorithms, such as Monte Carlo, that required fewer computational resources than exact algorithms, but have no guarantees for the optimality of the results. Regarding our starting protein, a NR able to metabolize, even incompletely, nitro compounds with experimental validation, the limited number of positions to design, and our computational resources, we chose to use a heuristic algorithm, and to enforce the resulting design results by MD simulation, allowing us to solve Newton’s equations of motion and thus to gather dynamical information. 

Second, the MD conclusion was based on the RMSD method. RMSD has the advantage of quickly translating into the stability of a protein. Nevertheless, it is essential to take this data with some hindsight. First of all, the main problem of the RMSD is closely related to the amplitude of error. Indeed, two identical structures could not be perfectly superimposed due to the movements of a single loop or a flexible terminus. In this case, such structures have a large global backbone RMSD and cannot be effectively overlapped by any algorithm that optimizes the global RMSD. The variations observed between our structure could be related to the high flexibility of H6, which is known to accommodate various substrates, in spite of their size. H6 is situated under FMNH2 and is connected to the rest of the protein through two loops. This unique situation allows, despite the mechanical rigidity of the helix, for a relative flexibility of the overall structure [37].

A similar approach has been performed in 2003. Loren et al. [38] computationally designed a receptor and a sensor protein with novel functions by using a DDE algorithm to construct efficient soluble receptors that binds TNT with high selectivity affinity. These designed receptors illustrate potential application of computational design and validate our approach.

Nonetheless, we only provided *in silico* results here. It would be interesting to produce the mutant and to proceed in vitro tests and calculations to get a better feedback on our work.

## 4. Materials and Methods 

### 4.1. Workflow

The protocol aims to generate mutants and to computationally validate our model by performing MD simulations aiming to provide information on the free binding energy and on the stability of our newly designed structure. Each run was repeated iteratively until we get satisfying results in terms of structure, stability, and affinity (Figure 3).

### 4.2. Computational protein engineering 

#### 4.2.1. Protein Preparation 

A 1.9 Å protein crystal structure (PDB ID: 5J8G [26]) of the bacterial NAD(P)H NR from *Enterobacter cloacae* in a ligand-bound complex with acid para nitrobenzoic (p-NBA) was obtained from the PDB [39] for use as a starting structure for computational modeling. This structure was examined and prepared for manipulation using the Structure Preparation feature in the Molecular Operating Environment software (MOE) [40]. Acid para nitro benzoic was removed and the structure was minimized using the Amber14 force field [41] to reach an energetically favorable conformation. Then all the unbound water and cofactor FMN molecules were removed from the structures.

#### 4.2.2. Cofactor Preparation 

FMNH2 structure file was built from 4PU0 [27] crystal structure and then converted into the appropriate conformer with MOE confsearch feature. FMNH2 was then superimposed to the oxidized FMN into the 5J8G crystal structure and the coordinates were converted into .mol files. The .mol files were then parameterized by Rosetta31 (molfile_to_params.py script) to produce a parameter file and a new .pdb file.

#### 4.2.3. Ligand Preparation 

HMX structure was downloaded from Chemspider [42]. Then, HMX was minimized and placed into the active site of the protein structure 5J8G, in superposition of the acid para nitro benzoic position and in a way that the nitro group to be reduced was at a distance of 3.8 Å from the N5 atom of the flavin. HMX was then converted into a .mol file using MOE. Partial charges were corrected using MOE features. The .mol file was then parameterized by Rosetta (molfile_to_params.py script) to produce a parameters file and a new .pdb file.

#### 4.2.4. Generation of HMX Rotamer Library 

An HMX rotamer resource file was generated by random sampling using the searchconf function in MOE. This rotamer library was completed with crystal structure of HMX retrieved from the Cambridge database [43]. The library was then used for all subsequent Coupled Moves protocol simulations that used HMX as substrate ligand.

#### 4.2.5. Resfile Generation 

A 4.5 Å space around HMX was then determined. All the amino acids with a lateral chain not facing the active site were removed. Amino acid involved in the stabilization of the HMX cofactor or in the catalytic activity of the protein were also removed. The remaining amino acid were put in a text file named “resfile.” This resfile gives information on which positions we want to design. Each target residue was allowed to mutate into rotamers of every amino acid (“ALLAA”).

#### 4.2.6. Design Method 

The computational protocol used in this study redesigns enzyme active site. Coupled Moves algorithm primarily focuses on the optimization of side-chains for ligand binding and allows for a certain backbone flexibility [44].

The NAD(P)H NR protein structure prepared in the Structure Preparation section was split into separate .pdb files, consisting of the apo-protein, the HMX substrate and the FMNH2 cofactor. Small molecule structure files (HMX and FMNH2) were converted into .mol files. The .mol files were then parameterized by Rosetta (molfile_to_params.py script) to produce a parameters file and a new .pdb file for each structure. Then, we combined the protein, HMX, and FMNH2 structure files into a single .pdb input file for coupled moves command.

The substrate parameters files were edited to include the path to the HMX rotamer library file described in the “Generation of HMX rotamer library” section. The resfile including all the positions to be designed was created. FMNH2 and the apo-protein were allowed to use sampling rotamers of their current identity. HMX was supplemented with the rotamer library previously generated. The design.sh script was used to call all the prepared input files. It contains the variables and instructions for the Coupled Moves method simulation run. 

#### 4.2.7. Model Evaluation (Data Analysis)

Low-energy sequences generated by the Coupled Moves protocol were discarded to remove redundant sequences across multiple protocol runs, with the lowest energy rotamer conformations saved for each unique sequence. Then, all the sequences were aligned, and the results were compiled to form a logo sequence. For each designed position, the overall height of the stack indicates the sequence conservation at that position, while the height of symbols within the stack indicates the relative frequency of each amino or nucleic acid at that position. The sequence with the most conserved amino acid at each designed position were retrieved and pushed through a molecular dynamics simulation.

### 4.3. Molecular Dynamics Simulations

#### 4.3.1. MD Methods

All the simulations were performed using the molecular dynamics program AMBER [41] using the Amber ff14SB force field for proteins, and the TIP3P model for all the water molecules in the system. The force field parameters for FMNH2 and HMX were provided by the General Amber Force Field (GAFF) [45]. Simulations were performed with the ligand bound to the protein with a cap comprising two layers of water molecules (TIP3PBOX) surrounding the complex within a distance up to 10 Å. The simulations systems were kept under isothermal/isobaric (NPT) conditions except for the heating phase.

Energy minimization was performed to obtain a low energy starting conformation for the subsequent MD simulation. The solvated complexes were minimized for a total of 5000 cycles, using the steepest descent method for 2500 cycles, followed by 2500 cycles of conjugate gradient. Then, a 1ns heating phase was performed from 0 to 300K at constant pressure and temperature. The equilibration/production was performed for 100 ns. Finally, the sampling phase was carried out at 300K for 30 ns. The time step of the simulations was 0.002 ps. Each MD simulation was performed in triplicate.

#### 4.3.2. Trajectory Analysis

The VMD software [46] was used to visualize trajectories generated during the simulation. Root Mean Square Deviation (RMSD) was used to determine structure stability. RMSDs were calculated for every simulation.

#### 4.3.3. Binding Free Energy Calculation

An ensemble average molecular mechanics energies combined with the generalized born and surface area continuum solvation (MM/GBSA) binding free-energy calculation was performed on the snapshots from the MD simulation to compare the binding affinity of HMX for the mutant and the WT NR. A total number of 200 snapshots were taken from the last 4 ns of the MD trajectory with an interval of 20 ps (only for all the trials where the ligand was successfully kept within the active state). The calculations were rendered by the MMPBSA.py [47] module of AMBER14. The MMGBSA method can be conceptually summarized as
∆G_MM/GBSA_ = G_complex_ − G_receptor_ − G_ligand_= ∆E_MM_ + ∆G_GB_ + ∆G_NP_ − T∆S
where ΔE_MM_ is the molecular mechanics interaction energy between the protein and the inhibitor, ΔG_GB_ and ΔG_NP_ are the electrostatic and nonpolar contributions to desolvation upon inhibitor binding, respectively, and –TΔS is the conformational entropy change. 

### 4.4. Docking Studies

Docking studies were performed with the MOE software. The last frame of the MD simulation involving the mutant NR complexed with HMX was extracted as .pdb. HMX, water molecules and ions were removed. The NADPH 3D conformer was downloaded from PubChem (PubChem CID:5884) and minimized in MOE using the AMBER14ff. A collection of poses was then generated from the pool of ligand conformations using one of the Triangle matcher placement methods. A total of 150 poses were generated. Each of the generated pose was attributed a London dG score. Poses generated by the placement methodology were then refined using the induced fit refinement scoring function GBVI/WSA dG with the generalized born solvation model (GBVI). Twenty-five top scoring poses were sorted out. The top five scores were then evaluated.

## 5. Conclusions

Explosives contamination has become a major environmental issue during the past few years. The pollution with energetic material started with WWII and is still going on, due to manufacturing industries, conflicts, military operations, armed forces training activities, dumping of munitions, etc. Different strategies have been studied to remediate contaminated sites: burning, bursting, or chemical destruction. However, such methods are either costly or environmentally damaging. Bioremediation has recently emerged as an alternative way to detoxify soils from HEMs by using bacteria or plant metabolic pathways. Still, the rate of detoxification is highly variable from one HEM to another. The high energy molecules HMX is particularly problematic, as its solubility in water is lower than the one of other HEMs like TNT or RDX. Degradation of HMX by bacteria has been observed for a few strains: *Methylobacterium, K. pneumonia, C. Bifermentans*, and *Phanerchaete chrysosporium*. However, the degradation of the molecule is either incomplete, questioning the toxicity of these intermediates of degradation or occurs at meager rates. 

To overcome these limitations, we rationally designed an enzyme known for its ability to reduce a broad range of nitro substituted compounds—the NR from *Enterobacter cloacae*. From structural data, we redesigned the active site specifically around HMX with the coupled moves algorithm of Rosetta. The mutated NR was then studied through MD simulations. Stability and affinity were calculated. HMX fits the designed active site in a better way than in the WT NR. The molecule makes more H-bonds, stabilizing the molecule, and exposing its nitro groups at a distance allowing for a hydride transfer from the FMNH2 cofactor. The distance remains acceptable for hydride transfer, and thus for the nitro reduction all along the 130 ns of the dynamics. Even if HMX is mobile in the active site, one of its eight oxygen atoms always remains close enough to the reductive N5 of the flavin to allow the hydride transfer. These results are encouraging, but further investigations need to be done. The basic functionality of the protein has to remain intact. Also, the total reduction reaction implies numerous HMX derivatives. Each of them has to be tested to check how far the reaction could go. Finally, it would be interesting to perform hydride quantum mechanics/molecular mechanics (QM/MM) studies to validate the reaction on an extended atomistic level.

## Figures and Tables

**Figure 1 ijms-20-05556-f001:**
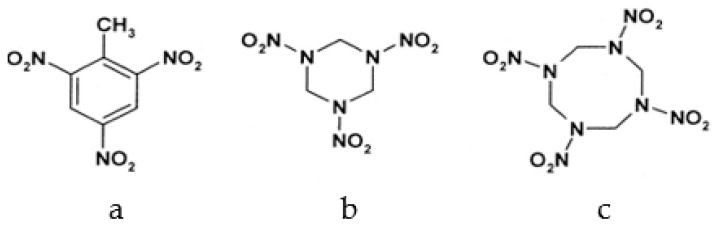
Two-dimensional chemical structures of (**a**) 2,4,6-trinitrotoluene (TNT), (**b**) hexahydro-1,3,5-trinitro-1,3,5-triazine (RDX), and (**c**) 1,3,5,7-tetranitro-1,3,5,7-tetrazocane (HMX).

**Figure 2 ijms-20-05556-f002:**
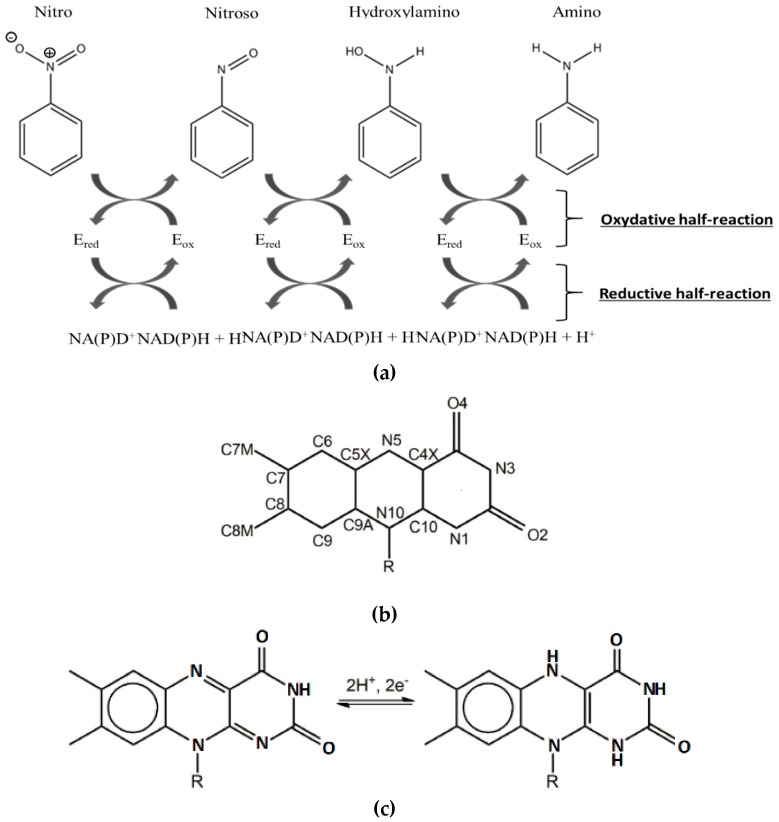
(**a**) Kinetic mechanism of the nitroreductase (NR) from *Enterobacter Cloacae*. NR is oxygen insensitive and involves two electron transfers at each reduction step. (**b**) Atom numbering of oxidized form of flavin mononucleotide (FMN)**.** (**c**) FMN and reduced form of flavin mononucleotide (FMNH2) structures.

**Figure 3 ijms-20-05556-f003:**
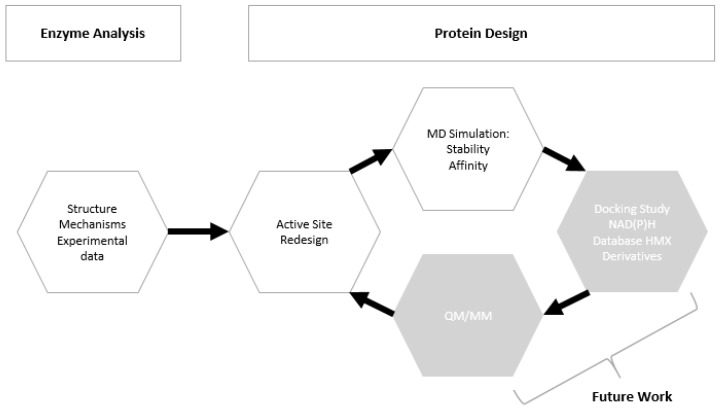
Pipeline/workflow of the study.

**Figure 4 ijms-20-05556-f004:**
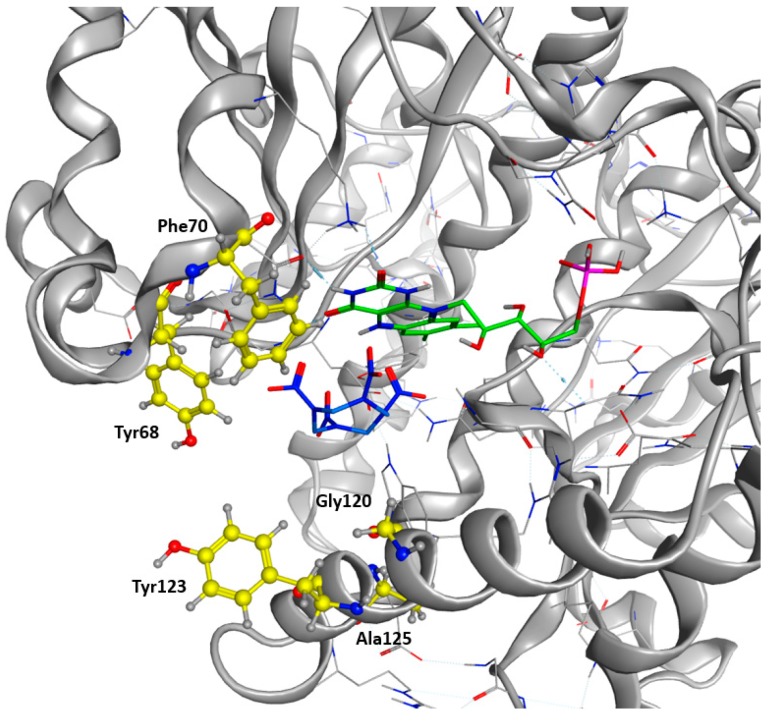
Target Residues for Design Trials. All the residues displayed in yellow are included in the resfile. FMNH2 cofactor is displayed in green, and HMX molecule in blue.

**Figure 5 ijms-20-05556-f005:**
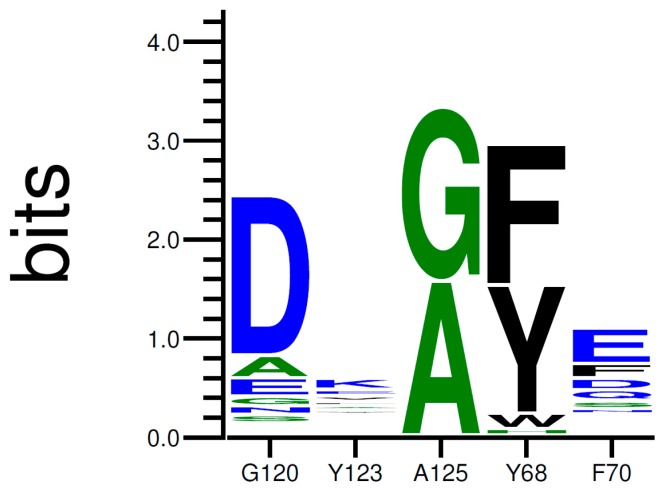
Relative Amino Acid Proportions at Positions 68, 70, 120, 123, and 125 in Low-Energy Structures. The relative size of each letter indicates their frequency in the sequences, and the total height of the letters shows the information content of the position, in bits.

**Figure 6 ijms-20-05556-f006:**
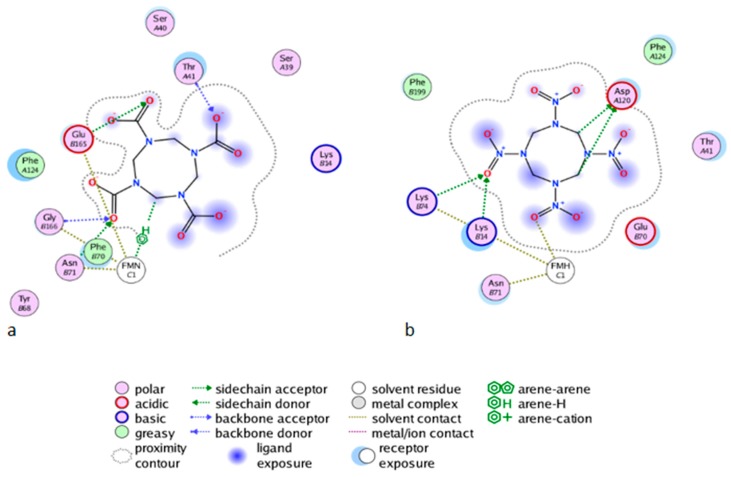
The Molecular Operating Environment software (MOE) Ligand Interactions application allows for the visualization of the protein active site in complex with HMX, in diagrammatic form. The diagram shows solvent interactions, H-bonds and surface of exposure. (**a**) Wild Type NR and HMX; (**b**) Mutant NR and HMX.

**Figure 7 ijms-20-05556-f007:**
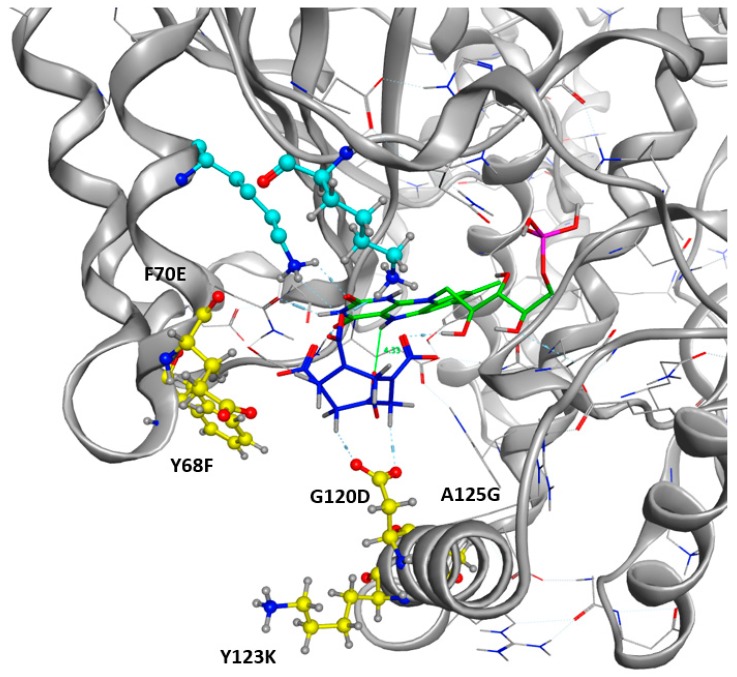
Mutated Residues after Design Trials. All the positions displayed in yellow were included in the resfile. FMNH2 cofactor is displayed in green and HMX molecule in blue. Non-mutated residues implied in H-bonds with the HMX are in cyan.

**Figure 8 ijms-20-05556-f008:**
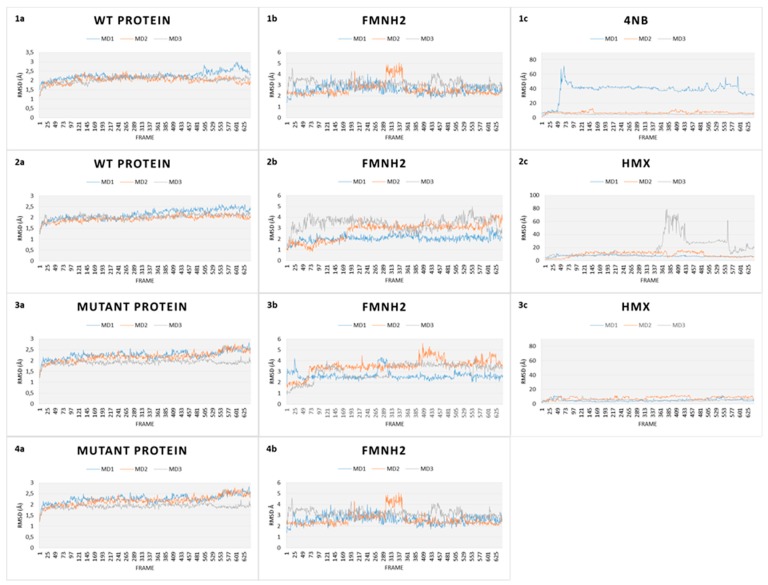
Root Mean Square Deviation (RMSD) plots over time of every simulation of both WT and mutant NR. (**1**) RMSD plots of WT protein in complex with 4NB. (**a**) Protein, (**b**) cofactor FMNH2, and (**c**) 4NB. (**2**) RMSD plots of WT protein in complex with HMX. (**a**) Protein, (**b**) cofactor FMNH2, and (**c**) HMX. (**3**) RMSD plots of mutant protein in complex with HMX. (**a**) Protein, (**b**) cofactor FMNH2, and (**c**) HMX. (**4**) RMSD plots of relaxed mutant protein without HMX (**a**) Protein, (**b**) cofactor FMNH2.

**Figure 9 ijms-20-05556-f009:**
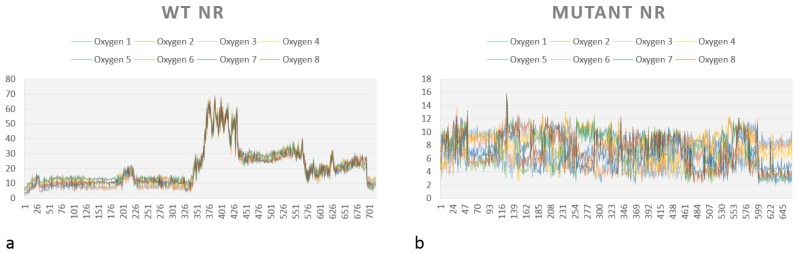
Average distance between each atom of oxygen from HMX and atom N5 of the FMNH2 in (**a**) the WT and (**b**) the mutant NR.

**Figure 10 ijms-20-05556-f010:**
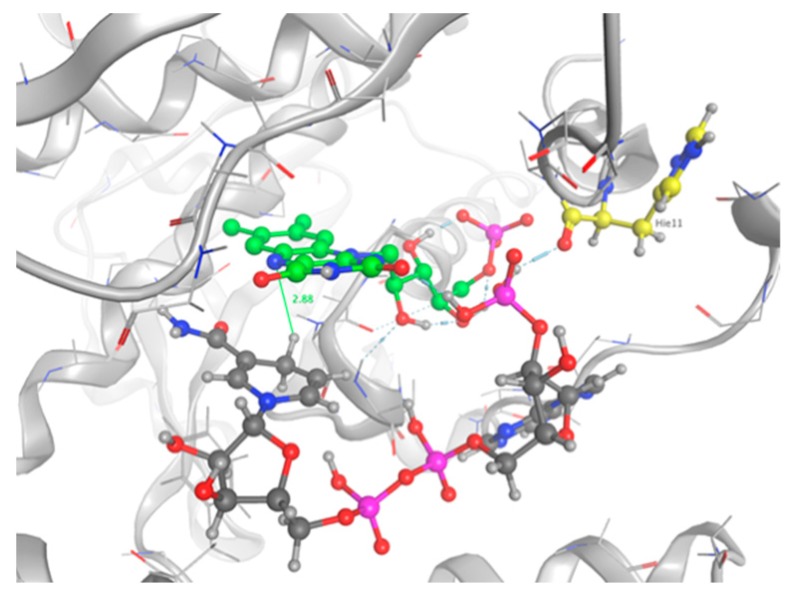
NADPH bound into the active site of the mutant NR. Protein backbone are shown in grey. Residue interacting with NADPH is shown as sticks. FMN is shown in green. The distance between the N5 of the flavin and the transferable H of NADPH is depicted.

**Table 1 ijms-20-05556-t001:** Table of free-energy calculation of NR/ligand complex. MMGBSA is calculated as a sum of a conformational energy terms supplemented with a solvation free-energy term calculated using continuum electrostatics.

Complex NR/HMX	mmGBSA (kcal/mol)
Wild Type NR/p-NBA	−15.2117
Wild Type NR/HMX	−11.8261
Mutant NR/HMX	−29.0704

**Table 2 ijms-20-05556-t002:** HMX derivative molecules.

HMX Derivative Molecule	Name
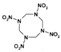	1,3,5,7-tetranitro-1,3,5,7-tetrazocane
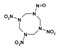	1,3,5-trinitro-7-nitroso-1,3,5,7-tetrazocane
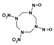	1,3-ditrinitro-5,7-dinitroso-1,3,5,7-tetrazocane
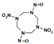	1,5-ditrinitro-3,7-dinitroso-1,3,5,7-tetrazocane
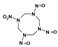	1-Nitro-3,5,7-trinitroso-1,3,5,7-tetrazocane
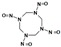	1,3,5,7-tetranitroso-1,3,5,7-tetrazocane
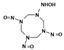	N-(3,5,7-trinitroso-1,3,5,7-tetrazocan-1-yl) hydroxylamine
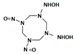	N-[3-(hydroxyamino)-5,7-dinitroso-1,3,5,7-tetrazocan-1-yl] hydroxylamine
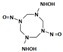	N-[5-(hydroxyamino)-3,7-dinitroso-1,3,5,7-tetrazocan-1-yl] hydroxylamine
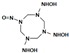	N-[(3,5-bis(hydroxyamino)-7-nitroso-1,3,5,7-tetrazocan-1-yl] hydroxylamine
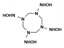	N-[3,5,7-tris(hydroxyamino)-1,3,5,7-tetrazocan-1-yl] hydroxylamine
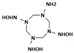	N1,N3,N5-trihydroxy-1,3,5,7-tetrazocane-1,3,5,7-tetramine
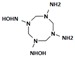	N1,N3-dihydroxy-1,3,5,7-tetrazocane-1,3,5,7-tetramine
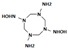	N1,N5-dihydroxy-1,3,5,7-tetrazocane-1,3,5,7-tetramine
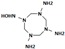	N1-hydroxy-1,3,5,7-tetrazocane-1,3,5,7-tetramine

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
