# Peer review of "Degradation of High Energy Materials Using Biological Reduction: A Rational Way to Reach Bioremediation"

_ijms, 2019, doi:10.3390/ijms20225556_

Round 1

Reviewer 1 Report

Aguero and Terreux describe their efforts in the design and in silico validation of a NAD-dependent Nitroreductase (NR) mutant that would eventually be proficient in the reduction of a highly recalcitrant explosive, namely HMX.

The authors identify a scientific problem of high interest, explosive bioremediation, and join this with the field of protein design, which is also a topic of evergrowing interest. I would consider this work positively in its intent, but I am disappointed by so many faults and oversights that I started thinking they could not be just typos, as they undermine the chemical soundness of the whole work.

Many of them have been specified in the pdf that the authors may gently find in the attachment in the form of comments to the main text. I will anyway summarize here some main concepts that convinced me in rejecting the work in the actual form:

1) In the method section I would expect to have much more details in order to let other scientists to repeat their procedures. In particular: (i) simulation and Rosetta parameters for substrates and cofactors should be provided and/or referenced, for comparison with previous literature and validation (ii) adopted scripts, if not referenced or if modified, should be provided. Moreover, I suggest a more scrupulous check of the details, as I hardly believe, for example,  that they equilibrated their MDs for 100 ns and produced only 30 ns, as their analysis has been accomplished on 130 ns in total, which I would expect to be the production time for this kind of systems (though people in the field is moving to much longer timeframes).

1a) Furthermore, manual positioning of the HMX molecule in the active site prevent other scientists in the field to recover the herein proposed results, as well as the outcome of the MDs. Have the authors evaluated any other binding mode? May they provide any statistics and/or divergence from the starting model of the binding mode? Have they considered to evaluate or cluster the outcome of the HMX binding in the MD, for further refinement of the design?

2) The manuscript should be deeply revised in term of its readability, several sentences are difficult to read, verbs are wrongly conjugated and sometimes plural numbers are mispelt. Moreover, figure labels and numberings are too small to be read, and sometimes the resolution is so poor that they are hardly interpretable. In particular RMSD and distance plots are not readable, and review is not possible.

3) Some of the identified mutations are conservative (A125G; Y68F), and the occurrence of the WT residues is comparable with respect to the selected residues. I suggest the authors to further validate these positions by evaluating their specific contribution both in the affinity measurements and in the MD simulations.

4) In the discussion paragraph, authors suggest that a further simulation of the redesigned NR is needed to test NAD(P)H binding and thus FMN reduction. I think that given the theoretical nature of the actual manuscript this kind of validation is highly needed. Ping-pong mechanism is a prerequisite for function for this class of enzymes, altered binding affinity may prevent activity.

Finally, I have to rise my personal criticism about the scientific impact in redesigning in silico such class of proteins. The authors are suggesting to alter only the substrate affinity of an enzyme that is already able to perform the reaction. Given the great wealth of knowledge raised in the last decade in the field of protein repurposing, several authors have shown that directed evolutionon is the elected and most appropriate technique to increase the catalytic efficiency for enzymes that already show reactivity to some extent. Whereas protein design is is highly recommended to implant a new function for a non-proficient protein or alter it completely, diverging from the natural one. The promiscuity observed for this class of enzymes and the wide scope of substrate tested in the literature, demonstrate that in this case in silico repurposing may represent the most difficult way to reach the final goal.

Reviewer 2 Report

High Energy Molecules (HEMs) are explosives, propellants, and pyrotechnics. They have been used for a wide range of purposes in the fields. Since they are causing environmental pollutions at various places, development of decontamination solutions is required, and biological degradation is regarded as promising method. Among HEMs, 1,3,5,7-tetranitro-1,3,5,7-tetrazocane (HMX) shows high resistance to chemical and biological degradation, due to its very low solubility. The authors design the nitroreductase from Enterobacter cloacae to facilitate HMX. The authors used the algorithm Coupled Moves from Rosetta to redesign the active site around the HMX. Then, they studied the newly designed protein by the Molecular dynamics simulations and affinity calculations.

The method was rational and well designed. HMX reaches a more stable state when complexed with the mutant enzyme. Affinity calculation also indicates an improved affinity.

However, it is not clear whether the designed enzyme can really function as a reductase on HMX. The authors raised the remaining problems and discussed in detail.

Because of the hydrophobic nature of HMX, it is not difficult to design high affinity structure. Thus, it requires experimental evidence for the enzymatic ability of the mutant enzyme.

Even if the mutant has enzymatic ability to degrade HMX, it seems difficult to obtain high degradation ability due to the low solubility of HMX.

The manuscript is sound as the preliminary study on the design of nitroreductase for HMX degradation. However, it needs further study to verify the activity of the mutant enzyme and to show the possibility for the practical application.

Reviewer 3 Report

In their paper, Aguero and Terreux present an interesting study, with a context of soils cleaning from HEM, coming from military uses and explosives since World War II. This study focuses on the design of a modified nitroreductase, in order to specifically bound the HMX molecule, difficult to eliminate because of its poor solubility and its stability. The paper deals with in silico approaches : computational design, MD simulations and their analysis.

The aim is quite interesting and the paper is quite pleasant to read, despite some inconsistencies. The figures must be improved in terms of quality, the resolution was actually too low and makes the understanding more difficult, especially Figure 5 and 7. However, I wasn't fully convinced by the results at the end and many improvements have to be done before possible publication in IJMS.

Major points:

-A major point concerns the design of the active site. I don't know in every details how the algorithm of Coupled Moves tools of Rosetta works, but it seems that there is presence of the ligand directly in the active site during the prediction of a modified active site. This has to be clearly precised in the methodology. If it is really the case, it includes a bias in the simulation to come. Indeed, it means that the positions of the mutated aminoacids are already oriented in the good way, toward the ligand. Thus, a comparison with a placement "by hand" of HMX in the wild type protein seems non relevant in that case. This has to be discuss. During the MD simulation of the mutant apo NR, what is the behavior of the mutated active site? Could the authors bring some insights in the article, section 2.2.1? Those information could help to discuss about this bias (of course if the algorithm works that way), but also to discuss about the apo mutant system (which appears only surprisingly during the RMSD analysis, not before).

-I understand the point regarding the position "in hand" of HMX within the cavity. But the problem is that, as the author says p. 8 "The assumption is that this variability, in terms of plasticity, is directly related to the wide range of substrate accepted by NR". In that case, how authors have validated the fact that the protein conformation associated to the p-NBA is correct for the placement of HMX? Precisions have to be added to the paper.

-Regarding the docking, it is said in the discussion that the derivatives of HMX has to be docked in the active site, but the problem of the helix 6, mentioned as the reason to not perform docking of HMX, is not mentioned again. If the docking is possible, why excluding this method for the HMX itself ? At least some clues on how the docking could be done have to be added. This point needs to be clarified.

- Concerning the simulations, the global simulation time (130 ns, 100 ns for equilibration and 30 ns for production) is quite low. Does authors made some replicates on their simulation in order to improve their sampling? I'm not sure that definitive conclusions could be establish with only 1 simulation of 130 ns for each system. Furthermore, rare event, such as ligand exit, has been sampled for HMX in the wild type, but the RMSD in the first ns is quite low, even lower that in the mutant NR. Maybe in another simulation, HMX could stay in the active site (or not), for example, and maybe in an other replicate of HMX-mutant, HMX could exit the active site... Authors have to comment this aspect in the discussion. If it is possible, the better will be to realize a second replicate for each system, or increase the simulation time of the simulations already made. This will strengthen the conclusion of the paper.

-Finally, I have some issues on the MM-GBSA approach. Authors said that the calculation of the affinity has been made on the last 4 ns of their simulations. But they claim that HMX and p-NBA have left the wild-type active site in their respective simulations. Does it mean that MM-GBSA has been made on the unbound state for those two cases? If it is really the case, those energies could not be compare with the HMX-mutant system.

Minor issues:

-On which selection the RMSD have been computed? I haven't found the information in the methodology.

-The simulation itself is very short (production phase: 30 ns) while the equilibration is quite long (100 ns). Can author justify their choice? Usually, equilibration step is shorter and not considered as a part of the simulation because the system equilibrates after the heating phase.

-In the methodology, it is said that the time step for MD simulations is about 0.02ps... meaning 20 fs ? I think that is a mistake and it is 0.002 ps, or 2 fs.

-Optimum distance between N5 flavin and nitro group is about 3.8 A. But p6, l 218, authors say that 4.35 A allow also an hybrid transfer. What is the range of distance for which the transfer is possible?

-p6, l 209: In Figure 5b, the interaction of D120 with the hydrogens are not shown. It is done on figure 6. Regarding those interactions, do author check that the hydrogen atoms associated to the carbon atoms have sufficient partial charges to create an H-bond type interaction with the G120D and not other form of weak interaction?

-For Figure 1, I think that a global picture of the protein, defining the position of the helices and the active site could be fruitful for a better understanding and had to be added.

-I suggest the author to carefuly read their manuscript with an english writer. The english style is generally correct, but some typos, repetition and weird sentences exist all along the text (for example, inconsistencies with the name of acid para nitro-benzoic, see P14, l 455 and 458)

-Finally, QM/MM methodologies must be discussed, in my opinion, in the discussion part and not only at the end of the conclusion. The consideration of a QM part has great impact on the system and QM/MM-MD methods could shed light on new insights (distances, angles, dynamical behavior).

Round 2

Reviewer 1 Report

Authors thoroughly revised their draft, and I particularly appreciated the NAAD docking part. However, in my opinion, given the theoretical nature of the present manuscript, control MD and affinity analyses should be performed to shed light on the role of some conservative mutations.

Furthermore, though the authors have replotted the RMSD and distance plots, I still feel that the quality of the figures is still lacking. Lines are too thick and superimposed to be distinguished, horizontal axes are in frames, whereas all over the manuscript the results are discussed in terms of time, making the evaluation unnecessarily troublesome/complicated. Even vertical axes should be plotted on the same scale in each figure, in such a way to support fast comparison among the simulation results. Finally, the number of frames is not always the same... Is there something to hide in the missing frames?

Just few other typos:

line 290: should probably appear 'weak interaction'

line 409: 'H3 et H4' --> 'H3 and H4'

line 422: what does 'top five scoring function' is related to? Probably top five scoring poses? Top five scoring complexes/structures?

line 607: 'proton transfer' --> 'hydride transfer'

Author Response

Reviewers 1

“Authors thoroughly revised their draft, and I particularly appreciated the NAAD docking part. However, in my opinion, given the theoretical nature of the present manuscript, control MD and affinity analyses should be performed to shed light on the role of some conservative mutations.

Furthermore, though the authors have replotted the RMSD and distance plots, I still feel that the quality of the figures is still lacking. Lines are too thick and superimposed to be distinguished, horizontal axes are in frames, whereas all over the manuscript the results are discussed in terms of time, making the evaluation unnecessarily troublesome/complicated. Even vertical axes should be plotted on the same scale in each figure, in such a way to support fast comparison among the simulation results. Finally, the number of frames is not always the same... Is there something to hide in the missing frames?”

All the typos have been reviewed.

Regarding the RMDS plots, a high-quality version has been provided in the article and as supplementary data. In this version, lines are thinner. In the supplementary data version, vertical axes are similar among all the MD simulations (6Å maximum for the protein and FMNH2, 90Å for the ligands HMX and p-NBA). In the paper version, the vertical axes are plotted on different scales among all our figures. We chose to show how far the HMX could go out of the active site. We also wanted to offer high reading precision. Plotting them on the same scale would force us to hide the information to a certain extent.

All horizontal axis are in frames. The whole dynamic represents 650 frames. The number of frames (649) and the time (130ns) is the same for all the MD. The time step of the simulations was 0.002 ps. The sampling is 1/10 000. We have resampled the MD for VMD with a rate of 1/10. However, for greater clarity, we also indicated in the text of the manuscript the equivalency in frames when durations in ns were mentioned.

However, if you highly recommend to do it, we can modify the axes in the paper instead of doing it in the supplementary data.

Reviewer 2 Report

English language of the revised manuscript was modified well. Although this is still a preliminary study, I agree that this manuscript gives an insight to the protein researchers. Therefore, I recommend this manuscript for publication.

Author Response

Reviewers 2

“English language of the revised manuscript was modified well. Although this is still a preliminary study, I agree that this manuscript gives an insight to the protein researchers. Therefore, I recommend this manuscript for publication.”

We thank you for your recommendation.

We agree that this manuscript is a preliminary study. Further studies to verify the activity of the mutant enzyme and to show the possibility for the practical application are ongoing: QM/MM calculations are coming first, and in a close future the in vitro production of the mutant is to be launched.

Reviewer 3 Report

The authors have greatly improved and clarified the study. The clarification about the triplicate is quite reassuring and greatly strengthen the study and the impact of the results about the designed active site for HMX. The results of docking study of NADPH in the newly active site gives more credit to the designed cavity.

English style has been considerably improved. Some few edits must be made (for example, l422 "the top five scoring function were evaluated". I think that the top five poses have been evaluated, not five scoring function; weird sentence l493, etc...) before publication.

Just a minor issue:

1) Regarding the triplicates RMSD plots, it appears that HMX goes out in one simulation of HMX/NR WT, but not in the two others. Could authors give some details about that  (values of RMSD in the stable simulations, behavior of HMX) and a comparison with the RMSD values observed in the NR mutant (4-6 A) ? Are those values (quite high for a ligand) associated to the conformational changes explained at l 313? Is the HMX behavior similar between the two systems? this part needs to be more detailed.

Once those modifications have been made, the article could be accepted for publication in IJMS.

Author Response

Reviewers 3

“The authors have greatly improved and clarified the study. The clarification about the triplicate is quite reassuring and greatly strengthen the study and the impact of the results about the designed active site for HMX. The results of docking study of NADPH in the newly active site gives more credit to the designed cavity.

English style has been considerably improved. Some few edits must be made (for example, l422 "the top five scoring function were evaluated". I think that the top five poses have been evaluated, not five scoring function; weird sentence l493, etc...) before publication. “

All the typos have been reviewed.

“Just a minor issue:

1) Regarding the triplicates RMSD plots, it appears that HMX goes out in one simulation of HMX/NR WT, but not in the two others. Could authors give some details about that  (values of RMSD in the stable simulations, behavior of HMX) and a comparison with the RMSD values observed in the NR mutant (4-6 A) ? Are those values (quite high for a ligand) associated to the conformational changes explained at l 313? Is the HMX behavior similar between the two systems? this part needs to be more detailed.

Once those modifications have been made, the article could be accepted for publication in IJMS.

Regarding the triplicates RMSD plots, it appears that HMX goes out in one simulation of HMX/NR WT, but not in the two others. Could authors give some details about that  (values of RMSD in the stable simulations, behavior of HMX) and a comparison with the RMSD values observed in the NR mutant (4-6 A)”

HMX goes out of the active site in one simulation of the WT NR/HMX, but not in the two others.

Indeed, in the first one, even if HMX seems to be stable in the first few ns, it finally goes out, to the same extent as p-NBA. HMX exhibits a RMSD value of 18.73 ± 1.19 Å in the first simulation, whereas it gets lowers RMSD values for the two other ones (respectively 8.52 ± 3.57 Å and 7.21 ± 1.19 Å).

As a comparison, HMX RMSD values for the mutant NR were 6.41 ± 1.20 Å, 4.35 ± 1.97 Å and 5.11 ± 1.76 Å for the triplicate MD.

“Is the HMX behavior similar between the two systems?”

This behavior is not similar among the two NR. In the WT one, HMX struggles to find a stable position in the active site. In the first MD, residues F124 and K123 of H6 do not manage to block HMX in the pocket. As a result, HMX leaves the active site. In the two-second ones, due to a slightly different orientation of their lateral chain, residues F124 and Y123 block HMX from escaping the active site. As a consequence, HMX manages to establish transient H bonds with K41. However, these H bonds are not sufficient to stabilize HMX, which rolls in the pocket.

“Are those values (quite high for a ligand) associated to the conformational changes explained at l 313?”

In the mutant NR, HMX is more stable in the active site and is adequately positioned. The mutations Y123K and G120D (H6) stick HMX against the re face of the flavin, establishing H bonds with the nitro groups of HMX. Y123K mutation plays a crucial role in maintaining HMX in the active site: in the WT NR, Y123 didn’t manage to stabilize HMX through H bonds. As a consequence, HMX escapes or is pushed in the pocket without reaching a stable state. In the mutant NR, K123 catches and stabilizes HMX through H bonding when the molecule moves away from the active site. G120D mutation also provides better stabilization through H bonds between the nitro groups of HMX and its two oxygens.

F124 prevents HMX from leaving to the same extent as in the WT NR. Moreover, this optimal support is reinforced by K41, which forms H bonds bridges between the protein, the FMNH2 cofactor, and HMX. Finally, HMX is stabilized at the required distance to observe the hydride transfer.

These observations have been added to the article in section 2.2.2 Stability Studies.